# Obesity and Risk for Lymphoma: Possible Role of Leptin

**DOI:** 10.3390/ijms232415530

**Published:** 2022-12-08

**Authors:** Carlos Jiménez-Cortegana, Lourdes Hontecillas-Prieto, Daniel J. García-Domínguez, Fernando Zapata, Natalia Palazón-Carrión, María L. Sánchez-León, Malika Tami, Antonio Pérez-Pérez, Flora Sánchez-Jiménez, Teresa Vilariño-García, Luis de la Cruz-Merino, Víctor Sánchez-Margalet

**Affiliations:** 1Department of Medical Biochemistry and Molecular Biology, School of Medicine, Virgen Macarena University Hospital, University of Seville, 41009 Seville, Spain; 2Department of Radiation Oncology, Weill Cornell Medical College, New York, NY 10065, USA; 3Oncology Service, Department of Medicines, School of Medicine, Virgen Macarena University Hospital, University of Seville, 41009 Seville, Spain

**Keywords:** lymphoma, obesity, leptin, adipokines

## Abstract

Obesity, which is considered a pandemic due to its high prevalence, is a risk factor for many types of cancers, including lymphoma, through a variety of mechanisms by promoting an inflammatory state. Specifically, over the last few decades, obesity has been suggested not only to increase the risk of lymphoma but also to be associated with poor clinical outcomes and worse responses to different treatments for those diseases. Within the extensive range of proinflammatory mediators that adipose tissue releases, leptin has been demonstrated to be a key adipokine due to its pleotropic effects in many physiological systems and diseases. In this sense, different studies have analyzed leptin levels and leptin/leptin receptor expressions as a probable bridge between obesity and lymphomas. Since both obesity and lymphomas are prevalent pathophysiological conditions worldwide and their incidences have increased over the last few years, here we review the possible role of leptin as a promising proinflammatory mediator promoting lymphomas.

## 1. Introduction

Lymphomas are lymphoid neoplasms that manifest as solid tumor masses. There are two variants of this disease called Hodgkin lymphoma (HL) and non-Hodgkin lymphoma (NHL). HL constitutes about 8% of all malignant lymphoid neoplasms and involves mature B lymphocytes, which correspond to the so-called Reed–Sternberg (RS) cell, which represents less than 1% of the total tumor [1,2]. By contrast, NHL represents most of the cases of malignant lymphoid neoplasms and involves not only mature B cells but also T and NK lymphocytes. Specifically, diffuse large B-cell lymphoma (DLBCL) is by far the most frequent subtype of NHL [3].

The overall incidence of HL is relatively low (two or three per 100,000 individuals), although there are different incidence peaks depending on age. The highest peak is observed in adolescents and young adults (15–35 years old), and almost 70% of them have EBV-negative HL, whereas the two lowest peaks appear in children and elderly adults, with a low prevalence (~30%) of EBV-negative cases [4]. However, the mortality rate in elderly patients is higher than in their young counterparts [5], especially in the male population [6]. By contrast, the overall incidence of NHL depends on the location, which goes from three to four per 100,000 male individuals in Vietnam or India to 17–18 per 100,000 in Israel Jews on the male population [7]. Moreover, NHL incidence increases exponentially with age. For example, 9.3 per 100,000 people under 65 years and 91.5 per 100,000 people more than 65 years had NHL from 2007 to 2011 in the USA [8].

Different risk factors have been associated with lymphoma development and progression over the years, such as immune deficiencies, viral and bacterial infections, inherited polymorphisms, acquired genetic mutations, or chemical exposures [9]. Nevertheless, another risk factor that needs to be taken into consideration is obesity, which is a proinflammatory state characterized by excessive adipose tissue that could promote not only different conditions and diseases such as cardiovascular disease, including coronary artery disease and high blood pressure, insulin resistance or type 2 diabetes mellitus [10] but also a huge variety of cancers [11,12,13], including lymphomas [14,15,16]. Interestingly, obesity-derived inflammation is potentially driven by the action of certain adipokines, such as leptin [17].

Leptin is a 16 kDa adipocyte-derived hormone that was first predicted in murine models [18,19] and described years later as the product of the *obese (Ob) gene* [20]. Leptin is mainly characterized having pleiotropic effects due to the existence of different leptin receptors (known as LEPR-a, LEPR-b, LEPR-c, LEPR-d, LEPR-e, and LEPR-f) [21,22]. The most important receptor is LEPR-b, which can fully transduce signals into the cell to activate a set of signaling pathways such as the Janus kinase (JAK) 2/signal transducer and activator of transcription (STAT) 3, insulin receptor substrate (IRS)/phosphatidylinositol-3 kinase (PI3K), or Src homology 2 domain-containing phosphatase 2 (SHP2)/mitogen-activated protein kinase (MAPK) [23,24]. The signaling pathways are typical of a type I cytokine receptor [25], and in fact, the receptor is present in every cell of the innate and adaptative immunological system [26], and that is why leptin is considered to be the link between the immune and metabolic system [27].

Leptin is mainly involved in the central control of energy metabolism [28] and obesity [29] but also plays a key role in other physiological systems and diseases, including autoimmune diseases [30], human dental pulp [31], bone metabolism [32], or cardiovascular diseases [33]. Of note, our group previously reviewed the striking role of leptin in other conditions that involve female reproduction [34], pregnancy [35], gestational diabetes [36], non-alcoholic fatty liver disease [37], atopic dermatitis [38], and even cancer [39,40]. 

Due to the increased incidence rate of both obesity and lymphomas over the past few decades, as well as the promising role of leptin in bridging obesity and many malignancies such as cancer, the aim of this article is to review the relationship between those two prevalent diseases and elucidate the role of leptin underlying this association.

## 2. Obesity and Lymphoma

### 2.1. The Relationship between Obesity and Lymphoma

The association between obesity and lymphoma has been largely discussed over the years. Overall, it seems that the risk of lymphoma increases at a higher body mass index (BMI) [41,42,43,44,45,46,47], but some studies have found non-significant positive associations or no correlations between both diseases [48,49,50]. Probably, this may be caused not only because HL and NHL are heterogenous diseases that involve many histological subtypes but also due to other variables such as location, gender, or age, as previously explained. In line with this notion, Willet et al. (2008) concluded that there was no evidence to support obesity as a determinant parameter for all types of NHL combined, whereas Ingham et al. (2011) found that obesity was associated with the risk of HL and most types of NHL, excluding FL [51].

The relationship between obesity and lymphomas was first suggested almost 50 years ago in a case–control study that evaluated 50,000 male students with different diseases, including 45 cases of HL and 89 cases of NHL (considered as “other types of lymphomas”) between 1916 and 1950, who were followed until 1974. In the study, it was found that an increased HL risk was prevalent among students who were obese, smokers, and coffee drinkers [52], whereas leanness was a predisposing factor for NHL in the same cohort [53]. Specifically, the risk of HL has significantly increased in both obese men and women [41,42,43,47], having been elevated more than two times compared with normal-BMI (18.5–25 kg/m^2^) patients. Interestingly, every 5 kg/m^2^ gained in BMI has proved to increase the risk of HL by 40% in both genders [42,47]. The British population passage cohort study, which analyzed 5.82 million patients, showed that every 5 kg/m^2^ increased the risk of HL by 10% [54]. Healthy adult women (considered from 19 to 44 years old) with a lower BMI also had a significantly increased risk of HL, whereas this association has been inversed in their older counterparts, suggesting that body size and strenuous physical activity may be associated with HL risk (at least in the female population), through immunologic, infectious, or genetic mechanisms [43]. By contrast, only a few studies found no significant increases in the risk of HL considering BMI as a single variant [49,50] or including other factors such as age or tobacco [55].

Regarding NHL, it has been found that the risk of the disease is elevated not only in obese (BMI ≥30 kg/m^2^) patients but also in the population with severe obesity (BMI≥35 kg/m^2^) [50] regarding both women and men [56,57], although it has also been found that obesity promotes the risk of NHL in female patients compared with their male counterparts [58]. The million women study evaluated the incidence and mortality for 17 specific types of cancer, including NHL, and found that the increase of 10 kg/m^2^ in BMI increased the relative risk of the disease [44]. Similarly, different studies have found that a 5 kg/m^2^ increase from normal BMI in men and women was also associated with the increased risk of NHL [45,46,47,59]. Of note, some factors, such as age, ethnicity, smoking status, alcohol consumption, gender [50,60,61,62], and probably menopause [44], should be considered since they have been found to increase the overall risk of the disease in obese people. Regarding responses to treatments, salvage chemotherapy and high-dose chemotherapy following autologous hematopoietic stem cell transplantation have been successfully tested as promising treatments in patients with relapsed lymphoma [63,64]. However, obesity influences responses to treatment in those patients and may impair overall survival [16].

NHL comprises a huge variety of histologic diseases that are also linked to obesity, especially DLBCL [56,65], which is aggressive cancer and the most common type of NHL since it represents about 40% of new cases. Moreover, one-third of DLBCL patients develop relapsed/refractory disease with poor prognosis, although some treatments have been successfully tested in the last few years to improve their clinical outcomes [66,67,68]. Up to now, some systematic reviews have shown limited evidence regarding the influence of obesity over DLBCL [69,70], but many studies support this association [45,46,47,59,71,72,73] even when a higher BMI has not been associated with the risk of overall NHL [74].

Similar conclusions have been extracted for FL: the second most common subtype of NHL [75]. Although it has been demonstrated that neither obesity nor height, waist/hip ratio, and physical activity has been associated with the risk of FL [76], other studies have found non-significant, positive associations between both diseases [45,57] and a higher risk of death [77]. Moreover, obesity was considered a risk factor not only for lymphoma in the head and neck (considering that the increase of 1 kg/m^2^ could enhance the risk of the disease by 1.3 times) [78], but also for T-cell NHL [79]. Regarding age at diagnosis, obese individuals between 40 and 49 years may have an elevated risk of DLBCL and FL [57]. Another study found that obese individuals under 45 years could have a greater probability of developing DLBCL compared to FL, especially in the male population [80].

On the other hand, obesity has also been suggested as a protective element for the development of NHL [48], and the relationship between obesity and the risk of small lymphocytic lymphoma (SLL) is *de facto* controversial [57,76]. In this regard, it was proposed the term “obesity paradox” to explain that obese individuals could have a more favorable prognosis compared to their healthy or underweight counterparts. The obesity paradox not only has been especially associated with cardiovascular diseases [81,82,83] but also with cancer [84,85,86]. It could be related to the inadequate use of BMI to measure obesity since protective muscle mass may also contribute to BMI [87]. Moreover, cancer produces weight loss, and it depends on the stage of the disease, so the excess of adipose tissue may represent an energy store for the survival of patients [39]. In line with those notions, other parameters, such as the waist circumference (WC) and the A body shape index (ABSI), defined by WC/ [BMI (2/3) × height (1/2)], have been analyzed in the Malmö diet and cancer study as better predictors than BMI for the risk of different hematological malignancies, including lymphomas [88]. Additionally, Alberta’s tomorrow project suggested that central adiposity, measured by WC, may be a stronger predictor of total cancer risk than BMI since men with more than 102 cm of WC had a significantly increased risk of NHL and hematological cancer [89].

### 2.2. Molecular Mechanisms Underlying the Association between Obesity and Lymphomas

The relationship between obesity and cancer has been extensively described in the past few decades due to the proinflammatory state that promotes tumor cell proliferation, including in lymphomas, as shown in Figure 1. It has been demonstrated that an excess of adiposity can modulate the aggressiveness of Hodgkin Reed–Sternberg lymphoma cells though different mechanisms that involve hypertrophied adipocytes, adipose stem cells, angiogenesis, and the release of pro-tumoral adipokines [90]. However, some studies have shown limited power, especially in NHL subtypes, and robust analyses to determine the etiologic mechanisms should be carried out [91].

In this sense, we already know that one of the most important pathways to regulate inflammatory responses associated with obesity is NF-κB, whose activity has been found to increase in mice with a high-fat diet compared with their low-fat diet counterparts [92] and could also mediate tumor cell proliferation, survival, and angiogenesis through the expression of different target genes, including *TNFA*, *BCLXL*, or *BCL2*, among others [93]. In many lymphoid malignancies, including HL, DLBCL, mucosa-associated lymphoid tissue (MALT) lymphoma, primary effusion lymphoma, or adult T-cell lymphoma/leukemia (ATLL), NF-κB signaling is considered a common hallmark since it is involved in lymphoma survival and growth by inducing anti-apoptotic and pro-proliferative gene programs [94,95]. In fact, many therapeutic approaches have been tested by targeting the NF-κB signaling pathway, such as rituximab in combination with ibrutinib, thalidomide, or lenalidomide in MCL [96] or the small interfering RNA (siRNA) nanotherapy in ATLL [97].

In addition, adipose tissue is one of the main sources of pro-inflammatory mediators. The major cytokine released by adipocytes is interleukin (IL)-6 [98], which could increase the risk of different cancers in obese patients, such as breast, liver, prostate, colon, and esophagus cancers [99], and lymphomas [100]. IL-6 acts as a growth factor together with IL-10 in NHL [101,102] and has been demonstrated to be involved in the resistance of PI3K pathway-targeted treatments via STAT3 or STAT5 activation [103]. Specifically, IL-6 is an important survival factor in MCL [102], and its level has been correlated with prognosis in DLBCL [104]. Moreover, pleural effusion lymphoma cell growth has been inhibited by using human IL-6 antisense oligonucleotides [105].

Circulating IL-8 is another cytokine secreted by adipocytes [106] and was found to be significantly higher in obese subjects compared with the non-obese controls [107], increasing inflammation and associated with different types of lymphomas. In this line, elevated levels of IL-8 have been found in gastrointestinal FL and MALT lymphomas [108] as well as DLBCL cells, which finally recruit neutrophils producing APRIL: a factor that promotes the development of different types of tumors and has been associated with poor survival in DLCBL due to DNA methylation and acetylation [109]. Circulating IL-8 levels have also been associated with concomitant infections and have been positively correlated with neutrophil counts in cutaneous T-cell lymphoma patients [110]. Of note, both IL-6 and IL-8 can be released by tumor-associated macrophages [111], which are widely known to promote cancer progression and metastasis and, in turn, these cytokines could participate in the recruitment and expansion of MDSCs [112], which have been extensively associated with poor clinical outcomes in both HL and NHL due to their role in immune evasion and cancer progression through different mechanisms [113]. In this context, we have recently found increased circulating levels of MDSC in DLBCL that decreased in patients with >24 months of survival [68].

Another protein, the *monocyte chemoattractant protein* (MCP)-1 (mostly known as C–C Motif Chemokine Ligand 2, CCL2), was also found in high concentrations in the serum of obese individuals [107] and was overexpressed in patients with triple-negative breast cancer, leading to cancer progression and metastasis [114]. Specifically, MCP-1 could be involved in the migration and localization of FL cells [115] and, together with its receptor (CCR2), has been suggested as a good factor to better identify DLBCL patients with high-risk by the international prognostic index since the high expression of these proteins has been associated with poor overall survival and progression-free survival [116]. The high expression of MCP-1 has also been found in other types of DLBCL called primary central nervous system lymphomas [117].

IL-1 also promotes inflammation in obese individuals [118] and has been demonstrated to be up-regulated in a huge variety of tumors, such as breast, head, neck, colon, pancreas, lung, melanomas, and lymphomas [119,120]. In lymphomas, IL-1α may have anti-tumoral properties [121], whereas IL-1β has been shown to be expressed in HL cells from areas of tissue with active fibrosis, and the receptor IL-1R2 may contribute to local and systemic modulation in the disease [120]. In this sense, an IL-1 blockade has been proposed with different treatments, such as chimeric antigen receptor (CAR) T cells targeting CD19 in acute lymphoblastic leukemia or DLBCL [122].

By contrast, the tumor necrosis factor (TNF)-α is also secreted by adipose tissues, and its levels correlate with the degree of adiposity [123], but its role in cancer remains controversial [124,125,126]. However, the role of anti-TNF-α therapies in increasing the risk of lymphoma has been described and seems clear in patients with autoimmune diseases, such as inflammatory bowel disease [127] or rheumatoid arthritis [128,129].

## 3. Leptin and Lymphoma

### 3.1. Leptin Signaling in Lymphoma

The metabolic abnormalities associated with an excess of adipose tissue include biochemical alterations such as high levels of plasma triglycerides [130] or peripheral insulin resistance, which lead to increased levels of insulin and glucose [131]. Importantly, other factors closely involved in obesity have been described as promoters of many diseases in the last decades, such as the adipokines leptin or adiponectin. Leptin is known to activate and promote the proliferation of monocytes and lymphocytes by activating JAK-STAT, PI3K, and MAPK [132,133]. Leptin signaling also drives the activation of many oncogenic pathways leading to the increased proliferation, epithelial-mesenchymal transition, migration, and invasion of tumor cells [134]. Specifically, leptin signaling pathways can promote lymphomas (Figure 2). Leptin binds its receptor LEPR-b to transduce activation signals into cells via JAK2, which is phosphorylated together with Tyr^985^, Tyr^1077^, and Tyr^1138^. STAT3 proteins bind phospho-Tyr^1138^ and are phosphorylated and translocated into the nucleus of dimeric units, activating the transcription of their targeting genes and leading to a huge variety of lymphomas, including DLBCL, unclassifiable diseases with features between DLBCL and Burkitt lymphoma, mantle cell (MCL), NK/T-cell (NKTCL), peripheral T-cell (PTCL), anaplastic large cell (ALCL) or intestinal T-cell lymphomas, as well as HL [135,136]. One of its targeting genes, the suppressor of cytokine signaling (*SOCS)-3*, has been found to be highly expressed in FL and ALCL [137,138]. Similarly, STAT5 binds phospho-Tyr^1077^ and is translocated into the nucleus after its phosphorylation, thus promoting not only DLBCL, PTCL, MCL, or HL (as STAT3 signaling does) but also γδ-T-cell and lymphoblastic lymphomas [139,140,141,142,143,144].

Moreover, SHP2 binds to phospho-Tyr^985^ and promotes the activation of the MAPK pathway, although leptin can also activate MAPK signaling independent of SHP2. The protein SHP2 has been associated with ALCL [145,146], whereas MAPK activity impairs outcomes in DLBCL, pediatric-type nodal FL, and plasmablastic lymphoma [147,148,149]. The phosphorylation of JAK2 also promotes the PI3K/AKT/mTOR signaling pathway via IRS activation. The IRS proteins are a family of cytoplasmic adaptor proteins with important roles in cancer [150]. Regarding lymphomas, IRS-1 has been demonstrated to activate anaplastic lymphoma kinase (ALK), which is involved in ALCL [151], and IRS-4 could mediate the mitogenic signaling of LB cells: a murine pre-T-cell lymphoma [152]. The activation of the PI3K/protein kinase B (AKT)/mammalian target of the rapamycin (mTOR) pathway also plays a key role in lymphoma, and many signaling pathway inhibitors have been developed to treat FL, DLBCL, MCL, small lymphocytic, and T-cell NHL [153,154,155,156,157].

At the cellular level, leptin signaling favors Th1 responses by enhancing IL-2, interferon (IFN)-γ synthesis, and inhibiting IL-4 production, which suggests that this adipokine may alter T-cell responses toward a proinflammatory phenotype [25,158]. The recruitment of proinflammatory cytokines by leptin could regulate the production of adhesion molecules, such as the vascular cell adhesion molecule (VCAM)-1 and intercellular cell adhesion molecule (ICAM)-1 [159,160], that have been found to be highly increased in newly diagnosed lymphoma patients and correlate with tumor dissemination, the aggressiveness of the disease, and worse response to treatments [161,162,163,164]. 

Leptin also induces TNF-α in many settings [165,166,167]. Although its role in cancer remains controversial (as previously explained), TNF-α has been shown to play a key role in the pathogenesis of NHL [168] and may increase the risk of disease together with leptin, especially in FL [169] and DLBCL, through polymorphisms in the *TNF rs1800629G>A* gene [170]. Additionally, TNF-α levels were higher in lymphoma from children compared with their solid-tumor counterparts [171], which suggests the relevant role of this protein in lymphomas. IL-10 and IFN-γ released by leptin may be implicated in lymphomagenesis since their circulating levels were increased in patients with BMI ≥ 25 kg/m^2^ compared to individuals with a lower BMI [172]. Although IL-10 may be associated with a higher risk of NHL, especially FL, IFN-γ was not associated with that risk [169]. The risk of lymphoma in patients with a higher BMI could be also increased by the release of IL-6 via leptin signaling [172], but it still needs to be completely elucidated since other studies did not find this association [169]. Of note, blood glucose was suggested as a prognostic biomarker for TCL [173], and the human oocyte testis gene 1, an antigen whose disruption promotes aberrant glucose homeostasis and defective hormone secretion, has been shown to decrease levels of insulin and leptin in TCL-bearing mice [174].

### 3.2. Leptin and LEPR Genes in Lymphoma

Leptin has been suggested to promote immune dysfunctions regarding body weight regulation and NHL: mainly DLBCL and FL. Regarding gene expression, lymphomas are mainly characterized by mutations that involve genes, such as B-cell lymphoma (*bcl)-2* [175], *bcl-6* [176], *p15* and *p16* [177], *p53* [178], or *myc* [179], which have been widely considered as biomarkers of poor prognosis in those diseases [180,181,182,183,184,185,186]. Specifically, BCL-2 is an antiapoptotic protein that belongs to the BCL-2 family together with other proteins, including (but not limited to) CL-X_L_ and BCL-W, with antiapoptotic properties, as well as the proapoptotic BAX, BAK, or BID proteins [187]. Leptin signaling has been demonstrated to play a key role in B-cell homeostasis through the induction of Bcl-2 [188], which could increase the risk of different pathological conditions. Leptin has demonstrated the ability to inhibit apoptosis and induce cell cycle by elevating Bcl-2 and cyclin D1 in leptin-receptor-deficient (*db/db*) mice [188]. Similarly, the Bcl-2 protein expression was elevated in db/db mice with diabetes [189,190,191], Which may be predisposed to develop lymphoma [192]. This adipokine also decreased the apoptosis of myocardial cells in rats via bcl-2 [193] and reduced the apoptosis of beta cells at physiological concentrations in vitro by maintaining or up-regulating bcl-2 expression, which could promote non-insulin-dependent diabetes mellitus [194,195]. Additionally, mild maternal protein deprivation during lactation in rat pups could affect thymic homeostasis by increasing the activity of leptin, which improves the levels of BCL-2 and inhibits the apoptosis of thymocytes [196]. In human trophoblasts, leptin also prevents apoptosis when elicited with high temperatures by increasing the BCL-2/BAX ratio [197]. In cancer, the silencing of leptin in HeLa cells, a cervical cancer cell line, has reduced the expression of bcl-2 and, consequently, promotes apoptosis and inhibits cell proliferation, thus suggesting the probable role of leptin in the progression of cervical cancer [198]. Those notions are especially significant since NF-kB, STAT3, PI3K, and AKT pathways are activated in lymphoma cells via leptin/LEPR signaling [199,200,201] and improving bcl-2 expression.

Several studies have analyzed the role of leptin genes in lymphomas (Table 1). Single nucleotide polymorphisms (SNPs) in leptin genes *LEP 2548GA and LEP 2548AA* have been shown to increase the risk of FL compared with *LEP 2548GG* [80]. Specifically, genetic polymorphisms in *LEP 2548GA* have been significantly associated with NHL under the homozygous co-dominant model and additive genetic model in the Caucasian population rather than among Asians after analyzing almost 7000 cases and 8000 controls [202]. The positive associations between *LEP 2548GA* and the susceptibility of NHL were also found in another study, but without statistically significant differences [203]. Moreover, SNPs in *LEP 2548GA* have not been suggested to increase the risk of cutaneous T cell lymphomas (TCL) but may be involved in the pharmacogenetic of different treatments for this disease since patients with AG or GG genotypes (with lower plasma leptin levels) could better respond to topical steroids (male patients) and phototherapy (female counterparts) compared with AA patients [204].

However, results regarding SNPs in other leptin genes, such as *LEP A19G* (also known as *rs2167270*) and its receptor *LEPR Q223R*, remain inconsistent. Polymorphisms in the *LEP A19G* gene have been correlated with BMI and an increased risk of DLCL and FL [71], but accumulating evidence from recent years has revealed that SNPs in LEP A19G are associated with a decreased risk of DLBCL [170] and FL [203]. In line with this, meta-analyses have reported that genetic polymorphisms in the *LEP A19G* gene were associated with a lower risk (or even decrease in the risk) of NHL among Latin American individuals [205] and Asians, Caucasians and mixed populations [207]. Additionally, polymorphisms in the *LEP A19G* receptor, *LEPR Q223R*, could not increase the susceptibility of NHL [206]. Other leptin genes, such as LEP 19AA, could decrease that risk [80], whereas the leptin receptor gene *rs1327118 G>C* has not been associated with susceptibility to the disease [170].

By contrast, the ghrelin GHRL SNP allele for *GHRL 4427G>A* has been inversely correlated with the risk of NHL, especially DLCL [208]. GHLR and leptin are hormones that play antagonistic roles in controlling energy balance [209] by increasing and decreasing the levels of neuropeptide Y (NPY), respectively [210,211]. NPY is a powerful appetite stimulator that serves as an immune mediator by releasing and inhibiting proinflammatory cytokines [208]. The role of NPY in disease risk and progression remains unclear since it has been found that NPY genes may affect the risk of NHL, especially FL [208], but no significant changes in NPY levels after treatment have been revealed in patients with different types of cancer, including NHL [168].

### 3.3. Serum Leptin and LEPR Expression in Lymphoma

Similar to leptin genes, the relationship between the concentration of circulating leptin or LEPR expression and lymphomas has also been studied (Table 2). It has been demonstrated that leptin levels increase the risk of NHL in individuals with BMI ≥ 25 kg/m^2^ [172]. In addition, the phosphorylation of STAT3 and AKT via JAK2/STAT and PI3K/AKT signaling pathways has led immunohistochemical studies to reveal high expressions of LEPR, p-STAT3, and p-AKT in many DLBCL cases [201]. In line with this notion, leptin has been demonstrated to stimulate the proliferation of DLBCL cells and inhibit apoptosis via the PI3K/AKT signaling pathway in vitro, whereas the pretreatment of DLBCL cells with LEPR-specific siRNA or the inactivation of PI3K/AKT activity depleted these responses [200]. Likewise, leptin has increased the cell viability of CTL MOLT-3 cells by promoting the recruitment and expression of Glut1, and LEPR-siRNA, which inhibited those responses [212].

By contrast, it has also been shown that leptin levels not only undergo slight changes after treatment in NHL adult patients [168] but also could be negatively correlated with the international prognostic score in HL and with the international prognostic index in NHL [214], suggesting a paradoxical role of leptin that has been previously explained not only in cancer [40] but also in other settings [218,219,220]. Also, Bertolini et al. (1999) studied patients with NHL (mainly DLBCL and FL but also other types of lymphomas such as MCL, PTCL, ALCL, large granular NK-cell lymphoma, and extranodal marginal zone lymphoma of MALT), whose leptin levels were not only similar regardless of the outcome but were also not correlated with age, gender, or even-free survival [213].

### 3.4. Linking Leptin, Lymphoma, and Obesity

Most of the studies have analyzed the relationship between leptin and lymphoma, obesity and lymphoma, or obesity and leptin. Therefore, only a few studies have analyzed the possible associations among leptin, lymphoma, and obesity. Recently, leptin has been positively associated with BMI and NHL risk [172]. Also, patients who survived the Burkitt type, non-Burkitt, and lymphoblastic lymphomas not only had low leptin levels but also a normal/low BMI (19.5 ± 3.4 kg/m^2^) [221]. On the other hand, leptin levels have been positively correlated with BMI but not associated with lymphoma risk [214]. Similarly, relationships between leptin or BMI with HL or NHL were not found in pediatric patients [216]. A BMI ranging from underweight to healthy values in children newly diagnosed with HL or NHL has been positively correlated with leptin. Thus, leptin levels were low in those patients at diagnosis [215,217] but may significantly increase after remission and predict the response to treatment or progressive disease [217]. Regarding SNP, it has been found that leptin gene polymorphisms were independent of BMI and did not alter the risk of NHL [80].

## 4. Conclusions and Future Perspectives

Leptin is a pleiotropic hormone released by adipocytes and not only plays critical functions in energy metabolism or appetite regulation but also takes part in multiple immune actions, including those that promote diseases such as cancer. At least in part, it seems that leptin may play a key role by increasing the risk of NHL or driving their progression, thus being associated with poor outcomes. However, leptin action in HL has been poorly studied and still needs to be completely elucidated. Also, we need to consider that adipose tissue secretes not only leptin but also other pro-inflammatory mediators that may take part in the development and progression of these diseases, such as IL-1, IL-6, IL-8, or MCP-1, as explained above.

Many factors may be implied to promote variations in leptin or LEPR expression in lymphomas, such as gene mutations since lymphomas are heterogeneous diseases with a huge variety of gene expression profiles, especially DLBCL [222,223]. In line with this notion, different gene mutations, and even the different signaling pathways activated depend on the type of lymphoma (Figure 2) and may lead to the opposite roles of the same leptin/LEPR SNPs for different lymphomas, as shown in Table 1. For example, the SNP of the *LEP 2548GA* gene increases the risk of FL, but cutaneous T-cell lymphoma patients with the SNP of this gene may respond better to treatments (Table 1). In this sense, further research is needed to find out the possible links between gene mutations, activated signaling pathways, and the gain-of-function or loss-of-function of those *leptin/LEPR* SNPs.

Other factors such as age, gender, ethnicity, or location have been proven to vary the incidence of lymphomas [224,225,226,227,228]. Similarly, although BMI includes body weight and is widely used as a marker for the development of several diseases, it might not be a good indicator of obesity because body weight also depends on muscle mass; also, BMI does not consider the fat mass of different body sites [229]. However, other parameters that have been examined to measure adiposity are increasingly used, such as WC, the waist:hip ratio, and even the waist:height ratio [230], which should be used to measure central adiposity more accurately and may better elucidate the actual relationship between obesity and lymphomas.

As previously explained, the different results obtained by linking obesity, leptin, and lymphomas highlight the paradoxical role of leptin. However, the results obtained by El Demerdash et al. (2021) further support the probable importance that leptin may have as a bridge between these pathphysiological conditions [172]. In any case, further research is needed to better elucidate this question not only in lymphomas but also in other diseases.

Interestingly, the idea that leptin plays a key role in lymphomas may be reinforced because of the use of metreleptin in other diseases. Metreleptin is a human leptin analog for the treatment of metabolic pathologies, such as acquired generalized lipodystrophy (AGL) [231]. Although AGL is associated per se with a high risk of lymphoma [232], metreleptin may accelerate that risk since AGL patients have been reported to develop NHL during metreleptin treatment, including peripheral TCL and ALK-positive ALCL [233] and may also jeopardize the recurrence of lymphoma [234], although further research is needed in this sense to completely confirm this statement.

We believe that leptin could be a potential bridge between obesity and lymphomas based on the available literature regarding leptin and LEPR genes, serum leptin, and LEPR expression in these types of cancer, as well as the different leptin signaling cascades that promote lymphoma. Overall, even though there is much evidence to support the critical role of leptin in increasing the risk of disease or being associated with a worse outcome of lymphoma, further studies, especially controlled and intervention studies, are needed to finally conclude the role of leptin as a link between obesity and lymphoma.

## Figures and Tables

**Figure 1 ijms-23-15530-f001:**
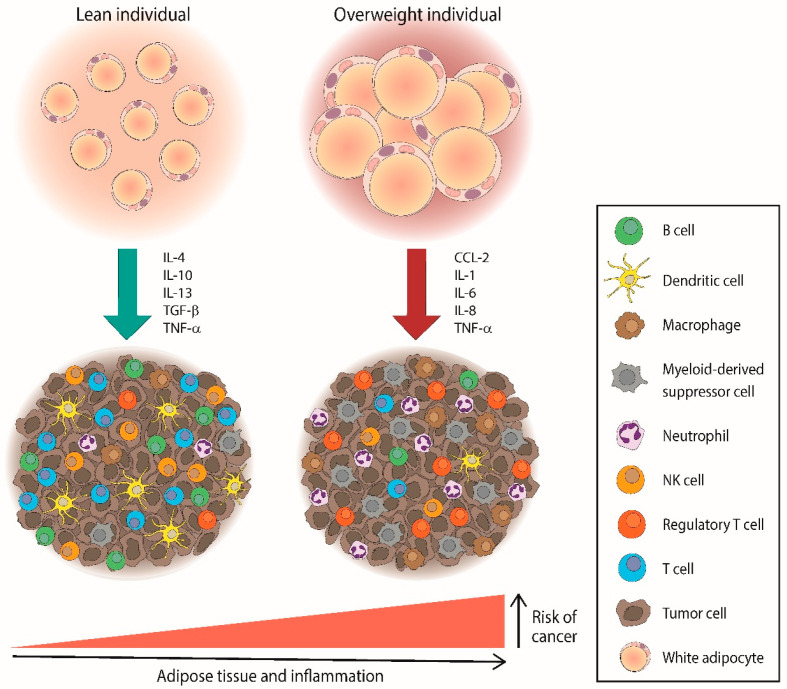
Overweight individuals could increase the risk of cancer, including lymphomas, through the accumulation of pro-inflammatory mediators and immunosuppressive cells within the tumor microenvironment. CCL—C–C motif chemokine ligand; IL—interleukin; TGF—tumor growth factor; TNF—tumor necrosis factor.

**Figure 2 ijms-23-15530-f002:**
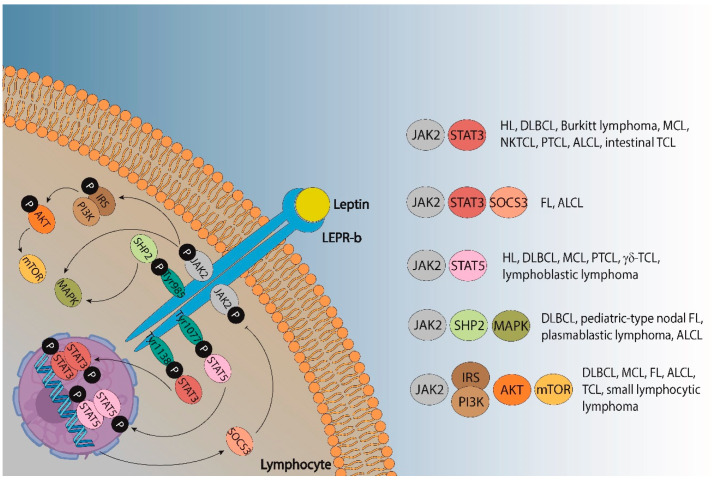
Leptin signaling pathways that could promote different types of lymphoma. ALCL—anaplastic large cell lymphoma; DLBCL—diffuse large B-cell lymphoma; FL—follicular lymphoma; HL—Hodgkin lymphoma; MCL—mantle cell lymphoma; NKTCL—natural killer/T-cell lymphoma; PTCL—peripheral T cell lymphoma; TCL—T-cell lymphoma.

**Table 1 ijms-23-15530-t001:** *Leptin/LEPR* gene polymorphisms analyzed in lymphomas. NHL—non-Hodgkin lymphoma; DLBCL—diffuse large B-cell lymphoma; FL—follicular lymphoma.

Reference	Type of Study	*Leptin/LEPR* Genes	Conclusions
[71]	Case-control	*LEP 19AG, LEP 2548GA, LEP 2548AA, and LEPR Q223R*	Polymorphism in *LEP 19AG* increased the risk of DLBCL and FL. Genetic interactions in *LEPR 223RR*, *LEP 2548GA*, or *LEP 2548AA* genes also increased the risk of NHL.
[80].	Case-control	*LEP 2548GA, LEP 2548AA,* LEP *19AA, and* LEPR *223Q>R*	Obesity was associated with risk of NHL, especially DLBCL. The risk of NHL was increased by *LEP 2548GA* and *LEP 2548AA* genes and decreased by *LEP 19AA*, particularly in men younger than in 45 years olds with FL. Conversely, no associations were found between lymphoma risk and *LEPR 223Q>R*.
[204]	Case-control	*LEP 2548GA*	Cutaneous T-cell lymphoma patients with leptin genes involving AG or GG genotypes may respond better to topical steroids and phototherapy.
[202]	Meta-analysis	*LEP 2548GA*	Gene polymorphism may increase the risk of NHL, particularly in the homozygote co-dominant model and the additive genetic model of Caucasian populations.
[205]	Meta-analysis	*LEP 19AG*	Gene polymorphism was associated with lower NHL risk under the homozygous codominant model, recessive genetic model (especially among the Latin American population), and additive genetic model.
[206]	Meta-analysis	*LEPR Q223R*	Gene polymorphism did not affect the risk of NHL, although it may be significantly increased in Asian and African individuals.
[203]	Meta-analysis	*LEP 2548GA, and LEP 19AG*	LEP 2548GA polymorphism increases NHL susceptibility and LEP 19AG is associated with a decreased risk of NHL, especially FL.
[207]	Meta-analysis	*LEP 19AG*	LEP 19AG may decrease the risk of NHL, especially in Asians, Caucasians, and mixed populations.
[170]	Case-control	*LEPR rs1327118G>C, and LEP rs2167270G>A (LEP 19AG)*	LEP rs2167270 G>A polymorphism was associated with the decreased risk of DLBCL in the recessive mode models among the Jordanian Arab population.

**Table 2 ijms-23-15530-t002:** Leptin/LEPR levels or expression analyzed in lymphomas.

Reference	Type of Study	Leptin/LEPR Levels or Expression	Conclusions
[213]	Case study	All patients: 23 (0–310) pg/mL. CR patients: 25 (0–310) pg/mL. PD patients: 21.5 (0–140) pg/mL.	Leptin levels were similar regardless of the response to treatment.
[168]	Case-control	Article not available *	After treatment, BMI, body weight and body fat mass decreased significantly. Also, low leptin levels were found before and after treatment compared with controls.
[214]	Case-control	Lymphoma patients: 16.4 ± 10.4 ng/mL. Controls: 10.3 ± 7.6 ng/mL.	Leptin levels were positively correlated with BMI but were not high in lymphoma patients at diagnosis.
[215]	Case-control	Patients: 6.0 ± 6.31 ng/mL. Controls: 5.9 ± 7.3 ng/mL.	There was no association between leptin levels and NHL in children.
[216]	Case-control	Patients: 8.2 ± 7.26 ng/mL. Controls: 7.5 ± 8.3 ng/mL.	There was no association between leptin levels and HL in children.
[200]	Case study	High LEPR expression in 39.8% of DLBCL patients	LEPR overexpression could be associated with DLBCL carcinogenesis via PI3K/AKT pathway. Also, leptin/LEPR signaling promoted the proliferation of DLBCL cells in vitro.
[217]	Case-control	Pre-treatment: 5.3 ± 1.56 ng/mL. Post-treatment: 9.8 ± 2.7 ng/mL. Controls: 6.7 ± 1.2 ng/mL.	Leptin levels were significantly lower in patients than in controls and increased in patients who achieved remission.
[169]	Case-control	Patients: 8.5 (3.8–17.1) ng/mL. Controls: 10.6 (5.2–21.8) ng/mL.	Serum leptin levels were significantly associated with NHL risk at diagnosis, but predicted a lower risk of FL.
[201]	Case-control	High LEPR expression in 45% of DLBCL patients	LEPR may promote JAK/STAT and PI3K/AKT signaling pathways and induce the phosphorylation of STAT3 and AKT, which may be involved in the prognosis of DLBCL.
[212]	Cases	Higher LEPR expression in tissues of T-cell lymphoma patients (58.3%) and in all cell lines, especially in MOLT-3 and Jurkat cell lines	LEPR overexpression was positively correlated with Glut1 expression. TCL MOLT-3 cell line demonstrated that leptin stimulated cell glucose uptake via promoting the recruitment and expression of Glut1.
[172]	Case-control	Patients: 4182.30 ± 246.95 pg/mL. Controls: 4782.00 ± 193.65 pg/mL	Leptin levels were significantly higher in women than in men and in obese patients compared with their non-obese counterparts, which increased the risk of NHL.

NHL—non-Hodgkin lymphoma; DLBCL—diffuse large B-cell lymphoma; FL—follicular lymphoma. * Article not available to check circulating leptin levels. The rest of the information in this study was taken from the abstract.

## Data Availability

Data are available at reasonable request.

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
