# Peer review of "Obesity and Risk for Lymphoma: Possible Role of Leptin"

_ijms, 2022, doi:10.3390/ijms232415530_

Round 1
Reviewer 1 Report
Jiménez-Cortegana et al. aimed to review the role of leptin in obesity-associated risk of lymphoma. They concluded that leptin might act as a pro-inflammatory mediator to promote the occurrence of lymphoma. From their review, this statement appears to be unconvincing and inconclusive. But, this can be improved. First, the literatures listed in Table 2 are inconsistent in supporting the link between serum leptin levels to any type of EBV-negative lymphoma. Moreover, there are altered expression of multiple adipokines and adipose tissue-derived factors, but the authors only single out leptin as a causal factor to lymphoma without solid literature evidence. In terms of chronic low-grade inflammation in obese subjects, the authors also fail to address this issue in an overview especially on the anti-inflammatory mediators in adipose tissue. Findings from leptin and leptin receptor knockout mice should be included in tumorigenesis. Second, the human SNPs of leptin/leptin receptor in Table 1 is interesting, but the findings are inconsistent with the same SNP. Moreover, gain-of-function or loss-of-function of these SNPs and their effects on the receptor signaling were not addressed to help understanding the pathogenesis. Third, chronic inflammation resulting from obesity is associated with multiple cancers. The authors mentioned cytokines secreted from immune cells, but this should not be confused with the contribution of adipose tissue where leptin is predominantly produced. In addition, many lymphoma has a strong link to gene mutation. The authors fail to exclude or address this issue.
Author Response
We appreciate Reviewer 1’s critical suggestions to further improve the manuscript. Based on the comments:
- The Reviewer is right since some studies do not correlate leptin with lymphoma development and progression positively. For this reason, we have rewritten some statements of the conclusion section, suggesting that leptin may be a potential link between obesity and lymphoma. Of course, there are many other variables involved: (1) adipose tissue releases a variety of mediators that can directly or indirectly promote lymphoma development (sometimes via leptin), (2) lymphomas are diseases characterized by a variety of gene mutations (some of them related to leptin), (3) other variable such as age, sex, or ethnicity may condition the development of lymphoma. With this, we mean that leptin is not the only factor to be considered as a link between lymphoma and obesity but, based on the majority of studies, leptin may be considered as another potential, promising biomarker. We have discussed these issues in the conclusion section.
- We already mentioned and explained a variety of adipose tissue-derived factors that may promote the risk of lymphoma, including IL-6, IL-8, MCP-1, or IL-1. They are found in section “2.2. Molecular mechanisms underlying the association between obesity and lymphomas”, paragraphs 3, 4, 5, and 6. Therefore, we consider that we did not single out leptin as a causal factor in lymphoma. To make it clearer, we have pointed it out in the conclusion section with the sentence “Also, we need to consider that adipose tissue secretes not only leptin, but also other pro-inflammatory mediators that may take part in the development and progression of lymphomas, such as IL-1, IL-6, IL-8, or MCP-1, as explained above”.
- The reviewer is right. Some studies regarding the role of leptin and Bcl-2 expression in leptin or leptin receptor-deficient mice has been cited in section “3.2. Leptin and LEPR genes in lymphoma”.
- As the Reviewer said, some findings are inconsistent with the same SNP of leptin/leptin receptor (Table 1). This point is very interesting because it may occur due to different gene mutations or signaling pathways that are activated in lymphomas. This could mean that the same SNPs of leptin/leptin receptor may have both anti- or pro-tumoral behaviors depending on the type of lymphoma. We have mentioned this issue in the second paragraph of the conclusion section, including one example from Table 1.
- We agree with the Reviewer that the gain-of-function or loss-of-function of these SNPs and their effects on the receptor signaling could help to understand the pathogenesis of different types of lymphomas. However, we did not address this issue because it is still unclear and further research is needed. Accordingly, we have mentioned it in the second paragraph of the conclusion section.
- We agree that cytokines from section “2.2. Molecular mechanisms underlying the association between obesity and lymphomas” can be released by immune cells, but the available literature reflects that those cytokines can be also released by adipocytes, including IL-6 (please see PMIDs 30001663 or 31980524), IL-8 (PMIDs 11238519, 12364441), IL-1 (PMID 12716739), or MCP-1 (PMIDs 35941819 or 30952712) that may ultimately contribute to both the release of leptin and/or lymphoma development and progression. Others, such as TNF-a, are not released by adipocytes but play a major role since they promote the release of leptin from adipose tissue (PMID 18037376). Therefore, TNF-a may indirectly induce the development of lymphoma via leptin.
- We consider that we did not exclude the link between lymphoma and gene mutations. In fact, in the third paragraph of the introduction section we mentioned that lymphoma development and progression is associated to acquired genetic mutations, among many other factors. Also, in the first paragraph of section “3.2. Leptin and LEPR genes in lymphoma” we already mentioned that “lymphomas are mainly characterized by mutations that involve genes such as B-cell lymphoma (bcl)-2, bcl-6, p15 and p16, p53, or MYC, which have been widely considered as biomarkers of poor prognosis in those diseases” and we addressed the implications of Bcl-2, also related to leptin in the disease. In addition, in the conclusion section, we already mentioned that “lymphomas are heterogeneous diseases with a huge variety of gene expression profiles, especially DLBCL”. However, we have rewritten the last sentence to allude to gene mutations.
Reviewer 2 Report
Jiménez-Cortegana and co-authors present a review in which they focus on the role of obesity in the development of lymphomas and analyze the possible role of leptin as a promoter of lymphoma.
The authors have done an excellent job; the review is well-structured and written. They reported all recent findings on the relationship between obesity and lymphoma and the mechanisms underlying this association.
The only note I would make to the authors concerns the two tables that are not easily usable by readers. Authors should find ways to make them simpler and less boring.
Author Response
We thank Reviewer 2 for the positive evaluation about the manuscript. We have improved both tables to make them simpler and less boring. In this sense, we have removed the “type of lymphoma” columns since almost all lymphomas from Table 1 and 2 are non-Hodgkin subtypes. Also, we have removed the number of patients/controls recruited from the “type of study” sections and simplified the conclusion columns.
Round 2
Reviewer 1 Report
Was the supplementary file entitled "Scheme 1. PCA score plots of differential metabolites of endometrium in dogs with pyometra and healthy dogs." uploaded incorrectly? Other than this, the authors have addressed my concerns.